# Peiminine Exerts Its Anti-Acne Effects by Regulating the NF-κB Pathway

**DOI:** 10.3390/antiox13010131

**Published:** 2024-01-22

**Authors:** So Jin Cha, Seon Sook Kim, Jin Hak Shin, Su Ryeon Seo

**Affiliations:** 1Department of Molecular Bioscience, College of Biomedical Science, Kangwon National University, Chuncheon 24341, Republic of Korea; soso52@kangwon.ac.kr (S.J.C.); wlsgkr1876@kangwon.ac.kr (J.H.S.); 2Institute of Life Science, Kangwon National University, Chuncheon 24341, Republic of Korea; painniche@kangwon.ac.kr; 3Institute of Bioscience and Biotechnology, Kangwon National University, Chuncheon 24341, Republic of Korea

**Keywords:** peiminine, *Cutibacterisum acnes*, inflammation, acne vulgaris

## Abstract

Peiminine is the main natural alkaloid compound extracted from the Chinese herb Fritillaria. Although peiminine is known for its antioxidant and anti-inflammatory effects in conditions such as mastitis and arthritis, its impact on inflammation induced by *Cutibacterisum acnes* (*C. acnes*) has not been explored. The aim of this study was to investigate the effect of peiminine on *C. acnes*-induced inflammatory responses in the skin and to identify the underlying mechanism involved. We discovered that peiminine inhibits the *C. acnes*-induced expression of inflammatory mediators such as pro-interleukin-1β (pro-IL-1β), cyclooxygenase-2 (COX-2), tumor necrosis factor-α (TNF-α), and interleukin-6 (IL-6) in mouse bone marrow-derived macrophages (BMDMs). Peiminine suppressed the activation of nuclear factor-kappa B (NF-κB) without affecting the activation of mitogen-activated protein kinase (MAPK) pathways such as JNK, ERK, and p38 MAPK. In addition, we found that peiminine suppressed inflammatory cytokine expression and ameliorated histological symptoms in *C. acnes*-induced mouse skin. Our study is the first to provide evidence that peiminine has an inhibitory effect on acne, and it points toward the potential of incorporating peiminine into cosmetic and pharmaceutical formulations for acne treatment.

## 1. Introduction

Acne vulgaris (AV) is a persistent inflammatory condition characterized by the accumulation of bacteria within hair follicles, resulting in increased sebum production, inflammation, changes in skin cell shedding, and the formation of clusters of bacteria in areas such as the face, chest, and back [1]. It is prevalent, affecting approximately 80% of adolescents and young adults [2], often leading to scarring, psychological stress, and a decreased quality of life [3].

A key culprit in acne vulgaris is the Gram-positive anaerobic bacterium known as *Cutibacterium acnes* (*C. acnes*). When skin pores are blocked, *C. acnes* proliferates and secretes lipases, breaking down neutral fats within the sebum and producing free fatty acids that trigger inflammation within hair follicles [4]. *C. acnes* is recognized by Toll-like receptor 2 (TLR2) [5], which plays a significant role in the innate immune response. The induction of inducible nitric oxide synthase (iNOS) and cyclooxygenase (COX-2) proteins leads to the generation of nitric oxide (NO) and prostaglandin E2 (PGE2), consequently activating reactive oxygen species (ROS)-related nuclear factor-kappa B (NF-κB) and AP-1 pathways [6]. NF-κB, a pivotal transcription factor in inflammation and immune responses [7], remains inactive in the cytoplasm but is bound to IκB. When *C. acnes* activates TLR2, IκB kinase (IKK) is activated, leading to the phosphorylation and degradation of IκBα. Once released, NF-κB undergoes translocation to the nucleus, where it induces the transcription of pro-IL-1β, COX-2, iNOS, and other pro-inflammatory cytokines [8]. *C. acnes* also activates the mitogen-activated protein kinase (MAPK) pathway, regulating both adaptive and innate immune functions and inducing the expression of pro-inflammatory cytokines such as interleukin (IL)-1β, TNF-α, and IL-6 [9]. In addition to TLRs, *C. acnes* is detected by pattern recognition receptors referred to as Nod-like receptors (NLRs). These receptors, including NLRP1, NLRP3, NLRC4, and AIM2, form multiprotein complexes of approximately 700 kDa called inflammasomes, along with the adapter protein apoptosis-associated speck-like protein containing a CARD (ASC) and pro-caspase-1. Inflammasomes trigger the activation of caspase-1, inducing the release of the pro-inflammatory cytokines IL-1β and IL-18 [10].

Historically, antibiotics such as clindamycin, erythromycin, and tetracyclines have been employed for acne treatment, but their extensive usage has contributed to antibiotic resistance [11]. Therefore, regulatory measures have been implemented in Europe since 2016 to limit the types of antibiotics used for acne treatment and the durations for which they are used, aiming to address antibiotic resistance issues [12]. Given the rise in antibiotic resistance, alternative methods to inhibit the proliferation of *C. acnes* in acne are being explored.

Peiminine, a major component of *Fritillaria*, which is a herb from traditional Chinese medicine, has been studied for its anti-oxidant and anti-inflammatory effects [13]. Previous research has indicated that *Fritillaria ussuriensis*, which originates from the bulbs of various species of *Fritillaria* plants, inhibits the production of inflammatory cytokines and MAPKs in mast cells [14]. Peiminine has also been found to inhibit mastitis induced by LPS through suppressing the AKT/NF-κB, ERK1/2, and p38 MAPK signaling pathways [15]. Peiminine has been reported for its ability to decrease the generation of ROS and mitigate the degeneration of dopaminergic neurons, thereby exhibiting a neuroprotective impact in models of Parkinson’s disease [16]. Our study investigated whether peiminine is effective as an anti-inflammatory agent against *C. acnes*-induced acne vulgaris in both in vitro and in vivo mouse acne models.

## 2. Materials and Methods

### 2.1. Materials

Peiminine was purchased from MedChem Express (Monmouth Junction, NJ, USA). ATP and Griess reagents were purchased from Sigma-Aldrich (St. Louis, MO, USA). The anti-phospho-NF-κB (p65) antibody was acquired from InvivoGen (San Diego, CA, USA). The anti-phospho-IκB, anti-IL-6, anti-MAPK, and anti-COX-2 antibodies were obtained from Cell Signaling Technology (Danvers, MA, USA). The anti-β-actin antibody was obtained from Santa Cruz Biotechnology (Dallas, TX, USA), and the anti-IL-1β antibody was obtained from R&D Systems (Minneapolis, MN, USA).

### 2.2. C. acnes

*C. acnes* (KCTC3314) was sourced from the Korean Culture Center of Microorganisms (Seoul, Republic of Korea). The cultivation of *C. acnes* involved the use of reinforced clostridial medium (Merck Millipore, Darmstadt, Germany) under anaerobic conditions, maintained using anaerobic Gas-Pak at 37 °C. After cultivation, *C. acnes* was subsequently centrifuged at 4500 rpm for 20 min at 4 °C, and the resulting bacterial pellets were thoroughly washed with PBS prior to utilization.

### 2.3. Cell Culture

In accordance with a previous method, bone-marrow-derived macrophages (BMDMs) were prepared [17]. Bone marrow progenitor cells were isolated from 8–12-week-old C57BL/6 mice and induced to differentiate into BMDMs by exposure to 30% L929 cell-conditioned medium (LCM). The culture medium for BMDMs was DMEM supplemented with 30% LCM, 10% heat-inactivated fetal bovine serum (FBS), and penicillin and streptomycin from Invitrogen (Carlsbad, CA, USA).

### 2.4. NO Aassay

The BMDMs were treated with peiminine for 30 min, followed by 6 h incubation with *C. acnes*. After 24 h, 50 μL of the culture supernatant was transferred to a 96-well plate and combined with the same volume of Griess reagent (Sigma-Aldrich, St. Louis, MO, USA). The resulting mixture underwent a 10 min reaction period, following which absorbance was measured at 540 nm.

### 2.5. Western Blot Analyses

BMDMs were seeded in 12-well plates at a density of 1 × 10^6^ cells/mL. The total cells were then resuspended in lysis buffer containing 1% Nonidet P-40, 1 mM EGTA, 50 mM Tris-Cl (pH 8.0), 1 mM NaCl, 10% glycerol, a protease inhibitor, 0.2 mM phenylmethylsulfonylfluoride (PMSF), 1 mM Na_3_VO_4_, and 10 mM NaF for 30 min on ice, after which the cell lysate was collected in a tube. For the analysis of secreted IL-1β, the BMDMs underwent priming with heat-killed *C. acnes* (1 × 10^6^ CFU/mL) for 6 h and were subsequently incubated with peiminine. The cells were then treated with ATP (5 mM) for 1 h to induce NLRP3 inflammasome activation. The culture supernatant was then collected in a tube and centrifuged at 1500 rpm for 3 min to eliminate any detached cells. The proteins in the lysate were separated using SDS‒PAGE and subsequently transferred to a polyvinylidene fluoride (PVDF) membrane. Following this, the membranes underwent blocking using TBS-T buffer (Tris-buffered saline with Tween 20) supplemented with 5% skim milk. The blocked membranes underwent overnight incubation at 4 °C with the primary antibody, followed by additional incubation with the proper secondary antibody for 1 h. The visualization of the bands was achieved using a chemiluminescence solution.

### 2.6. Reporter Gene Analysis

RAW264.7 cells were transfected with *Renilla*- and NF-κB-luciferase reporter constructs using Lipofectamine 3000 (Invitrogen, Carlsbad, CA, USA), after which the cells were cultured for 24 h. Following transfection, luciferase activity was subsequently measured using a Dual-Luciferase Assay System in accordance with the manufacturer’s protocol (Promega, Madison, WI, USA).

### 2.7. Quantitative Real-Time PCR (qRT-PCR)

Total RNA extraction was performed using TRIzol reagent (Invitrogen, Carlsbad, CA, USA). cDNA synthesis was subsequently carried out using ReverTra Ace qPCR RT Master Mix with gDNA Remover (TOYOBO, Osaka, Japan). The resulting cDNA was then subsequently amplified using SYBR Green real-time PCR Master Mix (TOYOBO, Osaka, Japan). The amplification process involved specific primers for the following target genes (Table 1). Each sample underwent triplicate analysis, and the mRNA expression levels were normalized to the reference gene β-actin. The obtained values were subjected to analysis using an AriaMX (Agilent, Santa Clara, CA, USA).

### 2.8. Mouse Acne Model In Vivo

BALB/c mice were obtained from Orient Bio Inc. (Seongnam, Republic of Korea) and subsequently raised at the Animal Laboratory Center of Kangwon National University. Moreover, the experimental procedures were performed in accordance with the guidelines and regulations specified by the Institutional Animal Care and Use Committee (IACUC, KW-201026-1) of Kangwon National University. The mice were divided into groups of three. Each mouse received an injection in the ear, either with *C. acnes* (1 × 10^8^ CFU per 20 µL in PBS) alone or in combination with peiminine (5 mg/kg). After 24 h, the mice were euthanized, and their ear tissues were harvested for subsequent analysis.

### 2.9. Histological Analysis

The collected ear tissues were fixed in 4% formalin solution and subsequently embedded in paraffin. Sections of the tissue, measuring 2–3 µm in thickness, were prepared and subjected to staining using hematoxylin and eosin (H&E). The stained sections were then examined via light microscopy using an Olympus microscope (Tokyo, Japan). Images of the observed pathological changes were captured by photographing the samples.

### 2.10. Statistics

The densitometric readings acquired from the Western blot analyses were assessed through the utilization of ImageJ 1.54g software (NIH, Bethesda, MD, USA). Data analyses were carried out utilizing GraphPad Prism 5.01 software (GraphPad software Inc., San Diego, CA, USA). The results are reported as the means ± standard deviations (SDs) derived from three independent experiments. Statistical comparisons between two groups were assessed using Student’s *t*-test. For analyses involving multiple groups, an analysis of variance (ANOVA) was conducted, followed by Bonferroni post hoc correction to further examine specific differences between the groups. * *p* < 0.05; ** *p* < 0.01; *** *p* < 0.001.

## 3. Results

### 3.1. Peiminine Inhibits Inflammation Induced by C. acnes in BMDMs

To explore the potential effect of peiminine on *C. acnes*-induced inflammatory signaling, we initially measured the cytotoxicity of peiminine in mouse BMDMs. The structural representation of peiminine is illustrated in Figure 1A. The BMDMs were exposed to increasing concentrations of peiminine for 24 h, after which cell viability was assessed via the MTT assay. Our findings indicated that after treatment with up to 120 μM peiminine for 24 h, cell viability did not significantly change (Figure 1B). Based on these results, subsequent experiments in the BMDMs were performed using 20, 60, or 120 μM peiminine. We analyzed the impact of peiminine on the mRNA expression of various *C. acnes*-induced inflammatory mediators through quantitative real-time PCR (qRT-PCR) analysis (Figure 1C–F). The results of the analysis indicated that pretreatment with peiminine led to a dose-dependent decrease in the mRNA levels of pro-IL-1β, COX2, IL-6, and TNF-α. We then monitored the effect of peiminine on *C. acnes*-induced inflammatory nitric oxide (NO) production (Figure 1G). As shown in Figure 1G, peiminine inhibited the generation of NO in a dose-dependent manner in the BMDMs. Taken together, these results indicate that peiminine inhibits the inflammatory signaling pathways induced by *C. acnes*.

### 3.2. Peiminine Suppresses the Protein Expression of Inflammatory Mediators Induced by C. acnes

We subsequently investigated the impact of peiminine on the protein expression of the pro-inflammatory mediators induced by *C. acnes*, including pro-IL-1β and COX-2, through Western blot analysis. As shown in Figure 2A,B, peiminine dose-dependently inhibited *C. acnes*-induced pro-IL-1β protein expression. The consistent reduction in COX-2 protein expression in response to peiminine further confirmed the anti-inflammatory influence of peiminine on signaling pathways triggered by *C. acnes* (Figure 2A,C).

### 3.3. Peiminine Inhibits NF-κB Activation Induced by C. acnes but Does Not Suppress MAPK Signaling

We then investigated the signaling cascades underlying the anti-inflammatory effects of peiminine triggered by *C. acnes*. Because *C. acnes* induces the expression of various inflammatory mediators through the activation of NF-κB and MAPKs (ERK, JNK, and p38 MAPK), we investigated whether the anti-inflammatory effects of peiminine occur through the inhibition of the NF-κB and MAPK signaling pathways. We assessed the activation of NF-κB and MAPKs via Western blot analysis, using antibodies specific to the phosphorylated forms of these proteins. When the BMDMs were treated with *C. acnes*, NF-κB phosphorylation was induced; however, in the presence of peiminine, NF-κB phosphorylation was notably inhibited, with the inhibition beginning at 60 μM (Figure 3A,B). We also observed the suppression of IκB phosphorylation by peiminine treatment (Figure 3C). However, peiminine did not affect the *C. acnes*-induced phosphorylation of MAPK (Figure 3A,D,E). To validate the inhibitory effect of peiminine on NF-κB activation triggered by *C. acnes*, we measured NF-κB-dependent gene transcriptional activation using luciferase reporter (Figure 3G). As depicted in Figure 3G, peiminine similarly restrained NF-κB-dependent gene transcription triggered by *C. acnes*. These results suggest that the suppressive influence of peiminine on *C. acnes*-induced inflammatory gene expression relies on NF-κB activation.

### 3.4. Peiminine Inhibits the Secretion of Active IL-1β

Pro-IL-1β undergoes conversion to its active IL-1β by an inflammasome complex and is subsequently released from cells. We subsequently examined whether peiminine inhibits the secretion of active IL-1β triggered by *C. acnes*. The BMDMs underwent a priming step with *C. acnes* and were subsequently subjected to treatment with ATP, a known activator of the inflammasome, to release active IL-1β. As shown in Figure 4A, we quantified the active IL-1β protein level in the culture supernatant using ELISA and observed that the presence of peiminine led to the concentration-dependent inhibition of IL-1β release. The decreased secretion of active IL-1β protein levels following treatment with peiminine was also observed by Western blot analysis (Figure 4B). These results indicate that peiminine decreases the secretion of active IL-1β from cells.

### 3.5. Peiminine Ameliorates C. acnes-Induced Inflammation In Vivo

Next, we explored the in vivo effects of peiminine by utilizing a mouse acne model. *C. acnes* with or without peiminine was inoculated into the ears of the mice. After 24 h, the ears inoculated with *C. acnes* exhibited prominent inflammatory symptoms, including erythema (skin redness) (Figure 5A). However, cotreatment with peiminine and *C. acnes* showed less erythema (Figure 5A). To further assess inflammatory responses, we monitored the histopathological changes in the ear tissues using hematoxylin and eosin (H&E) staining. When *C. acnes* was inoculated, inflammation and inflammatory cell infiltration in the dermis increased, but peiminine attenuated these reactions (Figure 5B). The ear thickness was consistently lowered in the peiminine-cotreated ear tissues (Figure 5C).

To assess the inhibitory effect of peiminine on *C. acnes*-induced ear inflammation, we quantified the mRNA levels in the ear tissues via qRT-PCR analysis. The levels of inflammatory markers such as IL-6, thymic stromal lymphopoietin (TSLP), COX-2, TNF-α, pro-IL-1β, and NLRP3 were elevated in the *C. acnes*-treated ear tissues, and the application of peiminine resulted in a decrease in their production (Figure 5D–I). To further assess the inhibitory effect of peiminine on *C. acnes*-induced ear inflammation, we investigated the protein expression of IL-1β in the ear tissues using Western blot analysis. The IL-1β protein levels in the ear tissues significantly increased after *C. acnes* injection, but peiminine treatment inhibited this increase (Figure 5J,K). Furthermore, we observed that the IκB phosphorylation level was lowered in the peiminine-cotreated acne ear tissues (Figure 5J,L). Overall, these findings indicate that peiminine improves *C. acnes-*induced acne symptoms by suppressing NF-κB signaling pathways in vivo (Figure 6).

## 4. Discussion

Acne vulgaris stands as a chronic inflammatory dermatological condition, exerting its impact on approximately 85% of individuals aged 12 to 24 years in the United States, and it is a prevalent skin disorder that affects 9.4% of the global population [18]. Moreover, approximately 40% of patients with acne experience long-term issues such as scarring and excessive pigmentation [19]. The pathophysiological factors of acne include changes in sebum production, excessive keratinization, inflammatory processes, and the proliferation of commensal bacterium *C. acnes* on the skin [20]. *C. acnes* constitutes one of the most prevalent skin microbiotas and plays a crucial role in the maintenance of a resilient and healthy skin barrier [21]. However, these cells proliferate when skin pores are obstructed, triggering inflammation within hair follicles [4].

Through exploring the effects of a *C. acnes* culture medium (acnes-CM) and formalin-inactivated *C. acnes* (F-acnes) derived from several distinct *C. acnes* strains, it was observed that both treatments significantly enhanced the intracellular accumulation of lipid droplets in hamster sebocytes. This increase was attributed to the upregulation of the de novo synthesis of triacylglycerols (TGs), observed both in vivo and in vitro [22]. *C. acnes* synthesizes a triglyceride lipase that elevates the concentrations of free fatty acids, notably palmitic and oleic acids [23]. The presence of palmitic acid, together with molecular patterns originating from *C. acnes* damage, activates TLR2, leading to the activation of inflammasomes and, subsequently, IL-1β signaling [24]. Conversely, oleic acid not only promotes the adhesion of *C. acnes* but also stimulates the proliferation of keratinocytes and the release of IL-1α [25].

For a long time, antibiotics such as erythromycin, tetracycline, clindamycin, and the vitamin A derivative isotretinoin were the mainstay treatments for acne. Isotretinoin exhibited anti-inflammatory effects by inhibiting the recognition of *C. acnes* by TLR2, thereby diminishing internal inflammatory responses. Nevertheless, systemic isotretinoin usage was linked to disruptions in both the gut and skin microbiota [19], which was particularly notable in females of childbearing age, raising concerns about the risks of birth defects and spontaneous abortion [26]. Moreover, antibiotics, which play a pivotal role in acne treatment, are now recognized for their ability to induce antibiotic resistance in *C. acnes* through chromosomal mutations or gene acquisition [27], consequently contributing to microbial imbalances [19]. The reported resistance to erythromycin and clindamycin underscores the necessity for a change in acne treatment strategies [27]. This emphasizes the importance of minimizing antibiotic usage and exploring alternative approaches to address this issue [19].

Peiminine is an alkaloid compound derived from the bulb of *Fritillaria thunbergii* Miq, a traditional herb in Chinese medicine [13]. The roots of *F. thunbergii* have various medicinal properties, such as cough relief, expectorant effects, and astringency [28]. Peiminine is known to have anti-inflammatory effects, influences cell apoptosis, and inhibits the proliferation of cancer cells [13,29]. The effect of peiminine is linked to the modulation of NF-κB activity and MAPK cascades. Peiminine treatment on osteoclasts inhibits the NF-κB and ERK1/2 signaling pathways, thereby regulating the differentiation and function of osteoclastogenesis and osteoporosis progression [13]. Peiminine treatment reduced both MAPK and NF-κB signaling in DNCB-induced atopic dermatitis skin tissue, suggesting its effectiveness in alleviating dermatitis symptoms by inhibiting these key inflammatory pathways [30]. Peiminine attenuated inflammation in an LPS-induced mouse mastitis model by inhibiting the phosphorylation of the NF-κB, ERK, and p38 signaling pathways [15]. In BV-2 cells, peiminine effectively reduced the expression of LPS-stimulated pro-inflammatory mediators such as TNF-α, IL-6, IL-1β, COX-2, and iNOS through the inhibition of the phosphorylation of ERK1/2, AKT, and NF-κB p65 [31]. In our study, we observed that peiminine inhibited the expression of pro-inflammatory cytokines mediated by *C. acnes* in the BMDMs by suppressing NF-κB transcriptional activation. However, the anti-inflammatory effect of peiminine on the BMDMs was not associated with MAPK signaling, indicating that peiminine exerts its anti-acne effect by controlling TLR2-activated NF-κB signaling pathways.

The inflammasome, a protein complex, is generally recognized to play a crucial role in inflammation and the pathogenesis of diseases such as Alzheimer’s, rheumatoid arthritis, type 2 diabetes, and autoimmune disorders to modulate inflammatory cytokine release [32]. NLRP3 inflammasome activation results in the maturation and release of inflammatory cytokines such as IL-1β and IL-18 [33]. *C. acnes* activates the NLRP3 inflammasome during acne development, resulting in a strong immune response. NLRP3 knockout mice have been reported to exhibit reduced inflammatory responses induced by *C. acnes* [34]. Our research demonstrated that *C. acnes* induces the activation of the NLRP3 inflammasome in BMDMs to release active IL-1β, and peiminine treatment effectively inhibits the production of active IL-1β.

Based on in vitro experimental findings, we studied the in vivo effects of peiminine using a mouse acne model. The inoculation of *C. acnes* into mouse ears induced characteristic symptoms of ear inflammation with skin redness, and peiminine treatment diminished this skin inflammation. It also reduced the expression of IL-1β, IL-6, NLRP3, TNF-α, TSLP, and COX-2, which are mediators of *C. acnes*-induced inflammation in mouse ears.

Nuclear factor erythroid-2-related factor 2 (Nrf2) plays a crucial roles as a regulator in response to oxidative stress and inflammation, and there is the existence of crosstalk between the NF-κB and Nrf2 pathways [35]. The absence of Nrf2 results in heightened NF-κB expression, contributing to an upregulation in the production of inflammatory factors. Conversely, NF-κB can influence the expression of downstream target genes by modulating the transcription and activity of Nrf2. Although our findings do not provide any evidence of peiminine exerting an effect on Nrf2 and NF-κB crosstalk, we aim to explore this relationship in our upcoming study.

## 5. Conclusions

In summary, our research revealed that peiminine decreases pro-inflammatory cytokine production induced by *C. acnes* by suppressing the NF-κB pathway both in vitro and in vivo. Our findings are the first to demonstrate that peiminine can control skin inflammation caused by *C. acnes*. These findings suggest that peiminine has the potential to serve as an alternative agent in the clinical treatment of acne. Considering its safety as a natural source, peiminine may be a valuable option for ameliorating symptoms of acne.

## Figures and Tables

**Figure 1 antioxidants-13-00131-f001:**
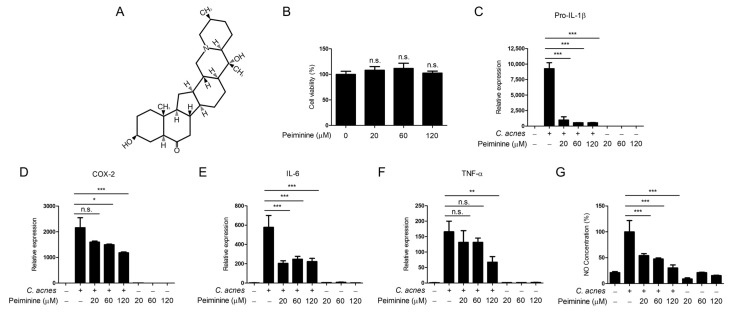
Peiminine inhibits *C. acnes*-induced inflammatory responses. (**A**) The chemical structure of peiminine. (**B**) Mouse BMDMs were exposed to various concentrations of peiminine (20 μM, 60 μM, or 120 μM) for 24 h, and cell viability was assessed using the MTT assay. (**C**–**F**) The BMDMs were pretreated with peiminine for 30 min and then exposed to heat-killed *C. acnes* (1 × 10^6^ CFU/mL) for 6 h. The mRNA levels of pro-IL-1β, COX-2, IL-6, and TNF-α were quantified via qPCR. (**G**) NO levels were measured using an NO assay. The results presented reflect the means ± standard deviations (SDs) obtained from three independent experiments. n.s. indicates a non-significant difference. * *p* < 0.05; ** *p* < 0.01; *** *p* < 0.001.

**Figure 2 antioxidants-13-00131-f002:**
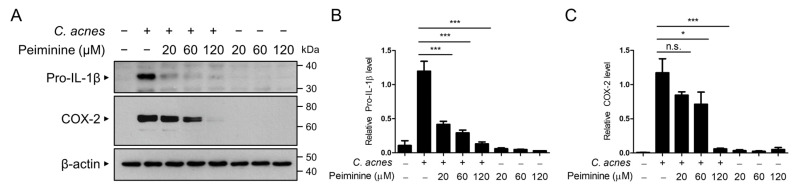
Peiminine inhibits *C. acnes*-induced pro-inflammatory protein expression. The BMDMs were pretreated with peiminine at concentrations of 20 μM, 60 μM, or 120 μM for 30 min. The cells were subsequently exposed to heat-killed *C. acnes* (1 × 10^6^ CFU/mL) for 6 h. The protein levels of pro-IL-1β, COX-2, and β-actin were assessed via Western blot analysis (**A**). The relative intensities of the protein bands were quantified (**B**,**C**). The results presented reflect the means ± standard deviations (SDs) obtained from three independent experiments. n.s. indicates a non-significant difference. * *p* < 0.05; *** *p* < 0.001.

**Figure 3 antioxidants-13-00131-f003:**
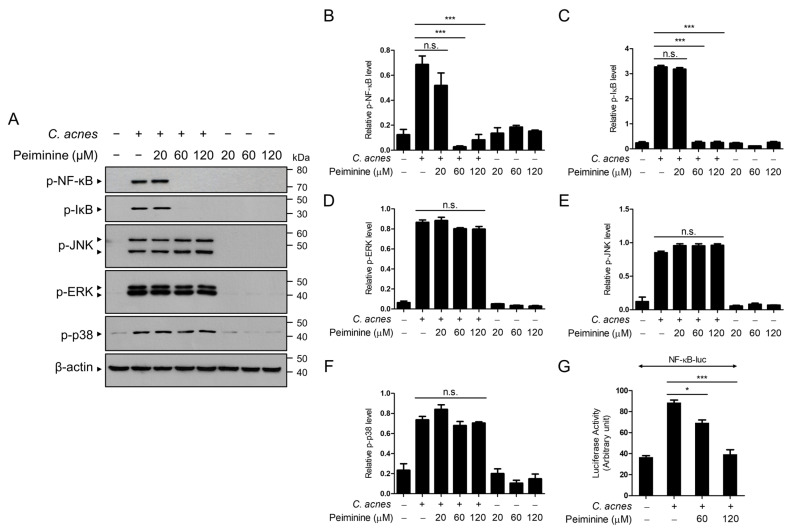
Peiminine inhibits the NF-κB transcriptional activation induced by *C. acnes*. (**A**–**F**) The BMDMs were exposed to peiminine at concentrations of 20 μM, 60 μM, or 120 μM for 30 min and subsequently treated with heat-killed *C. acnes* at a concentration of 1 × 10^6^ CFU/mL for 6 h. Western blot analysis was used to identify the protein levels of p-NF-κB (p65), p-IκB, and β-actin, after which the relative levels of the bands were quantified. (**G**) RAW264.7 cells were transfected and expressed with the NF-кB-luciferase reporter for 24 h. The cells were then treated with peiminine at concentrations of 60 μM or 120 μM for 30 min, followed by exposure to heat-killed *C. acnes* at a concentration of 1 × 10^6^ CFU/mL for 6 h. The results presented reflect the means ± standard deviations (SDs) obtained from three independent experiments. n.s. indicates a non-significant difference. * *p* < 0.05; *** *p* < 0.001.

**Figure 4 antioxidants-13-00131-f004:**
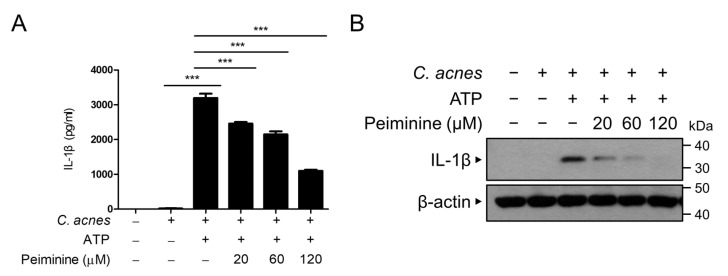
Peiminine inhibits the *C. acnes*-induced secretion of IL-1β. The BMDMs were initially primed with heat-killed *C. acnes* at a concentration of 1 × 10^6^ CFU/mL for 3 h. Subsequently, the cells were treated with 20 μM, 60 μM, or 120 μM peiminine for 30 min before being exposed to ATP (5 mM) for 1 h. The secreted IL-1β level in the culture supernatant was quantified via ELISA (**A**) and immunoblotted with anti-caspase-1, anti-IL-1β, and anti-NLRP3 antibodies (**B**). The results presented reflect the means ± standard deviations (SDs) obtained from three independent experiments. *** *p* < 0.001.

**Figure 5 antioxidants-13-00131-f005:**
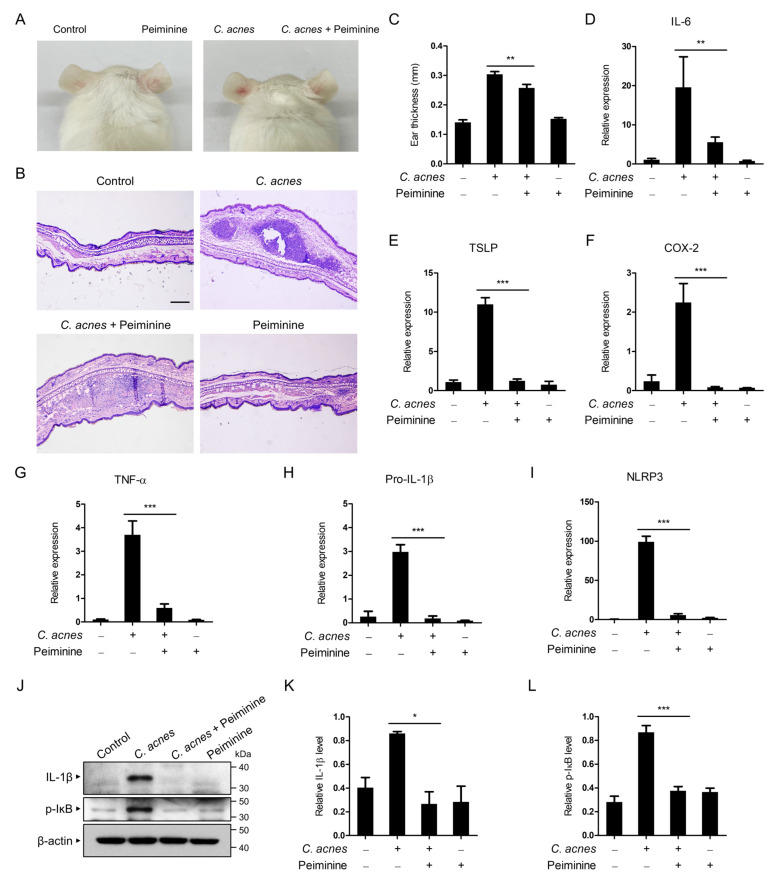
Peiminine inhibits mouse acne. (**A**) *C. acnes* (1 × 10^8^ CFU in 20 μL of PBS) was inoculated into the ears of mice either with or without peiminine treatment (5 mg/kg). Photographs of the ears were taken after 24 h. (**B**,**C**) The sectioned tissues were subjected to H&E staining, and the thickness of the ears was measured. The scale bar in the images represents 100 μm. (**D**–**I**) The mRNA levels in the ear tissues were measured via qPCR analysis. (**J**–**L**) Western blot analysis was performed using anti-IL-1β, p-IκB, and anti-β-actin antibodies, and the results were subsequently quantified. The presented results represent the means ± standard deviations (SDs) from three independent experiments. * *p* < 0.05; ** *p* < 0.01; *** *p* < 0.001.

**Figure 6 antioxidants-13-00131-f006:**
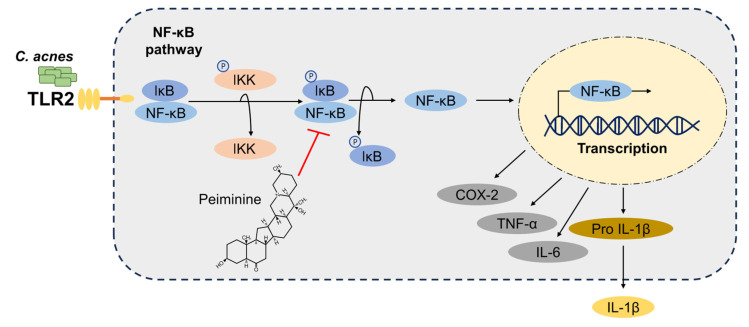
A graphical representation illustrating the role of peiminine in *C. acnes*-induced inflammation. Peiminine hinders NF-κB activation triggered by *C. acnes*, thereby preventing the secretion of IL-1β.

**Table 1 antioxidants-13-00131-t001:** The list of primer sequences used for qRT-PCR.

Gene	Primer Sequence	Accession No.	Tm (°C)	Amplification Efficiency (%)	Product Size (bp)
*Il1b*	F: GCCACCTTTTGACAGTGATGAG	NM_008361	84.5	102.2	165
	R: AGTGATACTGCCTGCCTGAAG				
*Il6*	F: TACCACTTCACAAGTCGGAGGC	NM_031168	83	95.7	116
	R: CTGCAAGTGCATCATCGTTGTTC				
*Tnf*	F: CCCTCACACTCACAAACCAC	NM_001278601	79.5	101.5	133
	R: ACAAGGTACAACCCATCGGC				
*Cox-2*	F: TTGGAGGCGAAGTGGGTTTT	NM_011198	85.5	92.2	148
	R: TGGGAGGCACTTGCATTGAT				
*Nlrp3*	F: GACCGTGAGGAAAGGACCAG	NM_145827	81.5	105.4	125
	R: GGCCAAAGAGGAATCGGACA				
*Tslp*	F: CCCTTCACTCCCCGACAAAA	NM_021367	80.5	98.6	61
	R: GCAGTGGTCATTGAGGGCTT				
*Actb*	F: AGAGGGAAATCGTGCGTGAC	NM_007393	85	103.3	138
	R: CGATAGTGATGACCTGACCGT				

## Data Availability

Data is contained within the article.

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
