# Peer review of "Peiminine Exerts Its Anti-Acne Effects by Regulating the NF-κB Pathway"

_antioxidants, 2024, doi:10.3390/antiox13010131_

Round 1

Reviewer 1 Report

Comments and Suggestions for Authors

Ms-antioxidants-2793443 needs minor revisions.

- The introductory part could be integrated by delving into the antioxidant aspects of Peiminine, in particular the mechanistic aspects.

-Nella the authors do not delve into the mechanisms of inflammasome activation, discussing only the aspects relating to the expression of NLRP3 without considering ASC and Casp1 which are part of the active complex.

- Information is available on mitochondrial oxidative stress possibly induced by C. Acnes.

- The possible modulation of NRF2 by NfkB could add an important aspect to the mechanism studied by the authors.

- Has the nuclear translocation of NfkB been quantified? Furthermore, are the entire NfkB or the p65 and p50 subunits indicated as translocated?

- The sequences of the primers used in Real-time PCR and indicated in 2.8 could be ordered in the table and completed by the identifying data co. me: T.A., n.cycles, length of the amplified pb, QPCR Amplification

Efficiency (%).

- The Blots presented in the figure lack the indication of the relative weight of the bands.

The manuscript must be rechecked as the text presents some typos

Comments on the Quality of English Language

Minor editing of the English language is required, the text presents some typos.

Author Response

The introductory part could be integrated by delving into the antioxidant aspects of Peiminine, in particular the mechanistic aspects.

A) According to the reviewer’s suggestion, reference to previous study on the antioxidant effect of peiminine was added to the introduction.

-Nella the authors do not delve into the mechanisms of inflammasome activation, discussing only the aspects relating to the expression of NLRP3 without considering ASC and Casp1 which are part of the active complex.

A) NF-kB is known to induce transcription of pro-IL-1b and NLRP3 during the priming step in the NLRP3 inflammasome signaling pathway. Therefore, the reduction in active IL-1b production by peiminine occurs through the ultimate impeding of the priming step of the NLRP3 inflammasome.

- Information is available on mitochondrial oxidative stress possibly induced by C. Acnes.

  1. A) Grange et al. showed that ROS production by keratinocytes upon  acnesstimulation through increased superoxide production acts via activation of NADPH oxidase (J Dermatol Sci 2009; 56: pp. 106-112). Although increases in ROS production by C. acnes were reported, the mitochondrial contribution of ROS elevation to P. acnes -induced inflammatory responses is still undefined.

- The possible modulation of NRF2 by NfkB could add an important aspect to the mechanism studied by the authors.

A) Nrf2 and NF-κB play crucial roles as regulators in response to oxidative stress and inflammation and there is an existence of crosstalk between these two pathways. The absence of Nrf2 results in heightened NF-κB expression, contributing to an upregulation in the production of inflammatory factors. Conversely, NF-κB can influence the expression of downstream target genes by modulating the transcription and activity of Nrf2. Although our findings did not provide any evidence of peimine on Nrf2 and NF-kB crosstalk, we aim to explore this relationship in our upcoming study in accordance with the recommendation from the reviewer.

- Has the nuclear translocation of NfkB been quantified? Furthermore, are the entire NfkB or the p65 and p50 subunits indicated as translocated?

A) We did not quantify the nuclear translocation of NF-kB, either for the entire NF-kB or the p65 and p50 subunits. Instead, we assessed NF-kB activation by utilizing a p65 phosphorylation-specific antibody in Western blot analysis. Furthermore, the translocation of NF-kB-dependent gene transcription was evaluated through a reporter assay in the Figure 3G. In accordance with the reviewer’s suggestion, it appears to be a promising approach to quantifiably assess the translocation of NF-kB. We will try to apply it in our next study.

- The sequences of the primers used in Real-time PCR and indicated in 2.8 could be ordered in the table and completed by the identifying data co. me: T.A., n.cycles, length of the amplified pb, QPCR Amplification Efficiency (%).

A) In response to the reviewer’s comments, the table has been made to include the sequences of the primers, gene number, products length, and qPCR amplification efficiency in the revised manuscript.

- The Blots presented in the figure lack the indication of the relative weight of the bands.

A) In response to the reviewer’s comment, we have indicated the relative molecular weight of protein was indicated in the revised figures.

The manuscript must be rechecked as the text presents some typos.

A) In response to the reviewer’s comment, we checked for typos throughout the manuscript.

Reviewer 2 Report

Comments and Suggestions for Authors

Peiminine is a major bioactive compound in the Chinese herb Fritillaria which has been known for its antioxidant and anti-inflammatory effects. This manuscript reported a study on its impact on inflammation induced by Cutibacterisum acnes (C. acnes) which is closely connected to acne vulgaris (AV). It showed that peiminine inhibited inflammatory cytokine expression and ameliorated histological symptoms in C. acnes-induced mouse skin, showing a potential of peiminine to be incorporated into cosmetic and pharmaceutical formulations for acne treatment. The experiments were designed well and the conclusions are clear. It is worthy to be published after minor revisions.

Practically, it might be difficult to use pure peiminine in cosmetic formulations because it is quite expensive. Alternatively, extract of herb Fritillaria is a good choice. And so I would like to recommend the authors to add an experiment on Fritillaria extracts. It revealed that peiminine suppressed the activation of nuclear factor-kappa B (NF-κB) but did not inhibit the activation of mitogen-activated protein kinase (MAPK). It is very interesting to invest whether the Fritillaria extracts has the suppressive effects on MAPK because it is a mixture of bioactives.

Author Response

Peiminine is a major bioactive compound in the Chinese herb Fritillaria which has been known for its antioxidant and anti-inflammatory effects. This manuscript reported a study on its impact on inflammation induced by Cutibacterisum acnes (C. acnes) which is closely connected to acne vulgaris (AV). It showed that peiminine inhibited inflammatory cytokine expression and ameliorated histological symptoms in C. acnes-induced mouse skin, showing a potential of peiminine to be incorporated into cosmetic and pharmaceutical formulations for acne treatment. The experiments were designed well and the conclusions are clear. It is worthy to be published after minor revisions.

Practically, it might be difficult to use pure peiminine in cosmetic formulations because it is quite expensive. Alternatively, extract of herb Fritillaria is a good choice. And so I would like to recommend the authors to add an experiment on Fritillaria extracts. It revealed that peiminine suppressed the activation of nuclear factor-kappa B (NF-κB) but did not inhibit the activation of mitogen-activated protein kinase (MAPK). It is very interesting to invest whether the Fritillaria extracts has the suppressive effects on MAPK because it is a mixture of bioactives.

--- Our research is conducted to identify the anti-inflammatory mechanism of peiminine on acne as a single compound. We concur with the reviewer’s suggestion regarding the the suppressive effects of Fritillaria extracts on MAPK, considering its economic viability. I plan to conduct an experiment to explore this aspect in the next time.  

Reviewer 3 Report

Comments and Suggestions for Authors

The manuscript entitled “Peiminine exerts its anti-acne effects by regulating the NF-κB pathway” is an interesting contribution to study the effect of Peiminine on anti-acne effects. The introduction is well written, objective is clear, material nad methods are well described, conclusions are supported by data, only minor details are needed, before accept the manuscript

Comments

Section 2.9

Groups of three mice is low, what about of the power of the experiment? Please discuss it

Discussion

Delete Lines 323 – 327

Author Response

The manuscript entitled “Peiminine exerts its anti-acne effects by regulating the NF-κB pathway” is an interesting contribution to study the effect of Peiminine on anti-acne effects. The introduction is well written, objective is clear, material nad methods are well described, conclusions are supported by data, only minor details are needed, before accept the manuscript

Comments

Section 2.9

Groups of three mice is low, what about of the power of the experiment? Please discuss it

 --- As the reviewer suggested, employing a larger number of animals would enhance the reliability of the data. Despite the use of only three animals in group in our experiment, it is important to note that the inflammation induced by acne bacteria does not lead to fatal toxicity or death. We believe that our sample size is adequate for conducting a meaningful analysis.

Discussion

Delete Lines 323 – 327

---Following the reviewer's recommendation, the discussion section has been added and reinforced.

Reviewer 4 Report

Comments and Suggestions for Authors

The researchers found that peiminine inhibits the expression of inflammatory mediators induced by C. acnes in mouse bone marrow-derived macrophages. Peiminine effectively suppressed the activation of nuclear factor-kappa B. Furthermore, the study revealed that peiminine reduced the expression of inflammatory cytokines and improved histological symptoms in C. acnes-induced mouse skin. These findings represent the initial evidence supporting the inhibitory effects of peiminine on acne, and the results appear to be compelling.

Author Response

Thank you for your comments